# Genome-Wide Analysis Reveals Chitinases as Putative Defense-Related Proteins Against Fungi in the Genomes of *Coffea arabica* and Its Progenitors

**DOI:** 10.3390/plants14203130

**Published:** 2025-10-10

**Authors:** Fernanda Rodrigues Silva, Mario Lucio V. de Resende, Katia V. Xavier, Jeremy T. Brawner, Mariana de Lima Santos

**Affiliations:** 1Department of Chemistry/Plant Pathology, Federal University of Lavras, Lavras 37200-900, MG, Brazil; mlucio@ufla.br (M.L.V.d.R.); santos-ml@outlook.com (M.d.L.S.); 2Department of Plant Pathology, Everglades Research and Education Center, University of Florida, Belle Glade, FL 33430, USA; kvianaxavier@ufl.edu; 3Department of Plant Pathology, University of Florida, 2550 Hull Road, Gainesville, FL 32611, USA; jeremybrawner@ufl.edu

**Keywords:** fungal resistance, *Coffea arabica*, *Coffea canephora*, *Coffea eugenioides*, molecular characterization

## Abstract

Chitinases have been demonstrated to enhance plant resistance to fungi in various pathosystems. Although there is evidence of the effectiveness of these proteins in coffee–fungus interactions, no genome-wide identification or characterization of coffee chitinases has been performed. In this study, we employed phylogenetic analysis, domain architecture, gene structure analysis, and subcellular localization to identify and characterize putative genes and proteins in the genomes of *Coffea arabica* and its progenitors, *Coffea canephora* and *Coffea eugenioides*. A total of 113, 47, and 69 putative chitinase proteins were identified in *C. arabica*, *C. canephora*, and *C. eugenioides*, respectively. These chitinases were classified according to their catalytic domains, GH18 and GH19, and into Classes I, II, III, IV, and V, as determined through phylogenetic analysis based on the *Arabidopsis thaliana* classification. Furthermore, based on orthologous analysis, we identified ten, six, and seven putative chitinases associated with fungal defense responses in *C. arabica*, *C. canephora*, and *C. eugenioides*, respectively. These findings are valuable for future studies focusing on coffee chitinases, particularly on genetic programs involved in plant pathogen resistance.

## 1. Introduction

Coffee is one of the most economically significant commodities globally. In the 2023/2024 season, total production reached 178.0 million bags [1]. Among *Coffea* species, *Coffea arabica* (Arabica coffee) and *Coffea canephora* are noteworthy, comprising 57.4% and 42.6% of global commercial coffee production, respectively [1]. *Coffea arabica* is allotetraploid (2*n* = 4*x* = 44), distinguishing it from other *Coffea* species that are diploid (2*n* = 2*x* = 22) [2]. It originated from a single natural hybridization event between *C. canephora* and *C. eugenioides* [3]. Despite being allotetraploid, this species has diploid meiotic behavior and follows disomic inheritance [4,5].

Overall, *C. arabica* is considered to have superior drinking quality compared to *C. canephora* due to its lower caffeine and higher sucrose content [6,7]. *Coffea arabica* is cultivated in tropical and subtropical regions across Africa, the Caribbean, Central America, Mexico, South America, Asia, and Oceania. Brazil is the largest producer of Arabica coffee, accounting for 56.27% of global production [8]. The cultivation of coffee is highly sensitive to climate conditions, posing significant challenges in tropical regions where climate change contributes to production losses, reduced yields, compromised quality, and an increased incidence of pests and diseases [9,10]. Various fungal diseases significantly impact coffee production worldwide [11]. Although coffee producers have adopted resistant cultivars to manage fungal diseases such as coffee leaf rust (*Hemileia vastatrix*) [12], resistant cultivars for other diseases, such as brown eye spot (*Cercospora coffeicola*), have yet to be developed [13]. In this context, studying plant defense mechanisms in coffee is particularly relevant, since fungal pathogens remain a major challenge to coffee production.

Understanding plant defense mechanisms is crucial in the pursuit of disease resistance. Currently, the plant’s immune system is described through two main lines of defense [14,15]. In the first, the recognition of molecules common to microorganisms (microbe- or pathogen-associated molecular patterns, MAMPs/PAMPs), such as fungal chitin, initiates PAMP-triggered immunity (PTI) [16]. The second line of plant defense is triggered by the recognition of effector molecules and is known as effector-triggered immunity (ETI). Effectors are typically molecules that enhance the virulence of pathogens [14]. Despite involving different ligands, PTI and ETI activate signaling pathways that are frequently overlapping and interconnected, forming a continuum of signaling [17,18]. These signaling events result in plants activating defense responses, including the production of reactive oxygen species (ROS), antimicrobial compounds, and pathogenesis-related proteins (PRs) such as antioxidant and hydrolytic enzymes [18,19].

Hydrolytic PRs, such as chitinases, exhibit constitutive activity and demonstrate heightened activity in response to biotic and abiotic stresses [20]. These proteins display high diversity and functionality, playing critical roles in various cellular processes such as growth, development, and stress responses, particularly in defense against phytopathogens [21]. Chitinases catalyze the hydrolysis of β-1,4 bonds in chitin [22], inhibiting pathogen growth and establishment. Moreover, by breaking down chitin, these enzymes release PAMPs, which are subsequently recognized by the plant’s immune system, thereby activating the defense response [19].

Chitinases are distributed across various plant chromosomes, and their classification is based on amino acid sequence similarity, three-dimensional structures, and catalytic mechanisms [23,24,25]. They are initially divided into two groups according to their catalytic domains: GH18 (Glyco_hydro_18) and GH19 (Glyco_hydro_19). Subsequently, they are categorized into five Classes (I, II, III, IV, and V) based on amino acid sequences, conserved motifs, and the presence of a chitin-binding domain (CBD) [26,27]. Most plant chitinases belong to Classes I, II, and IV, which are part of the GH19 family [20,22].

Class I proteins contain a CBD, a GH19 catalytic domain, and a C-terminal extension. Class II proteins are homologous to Class I but lack the CBD [28], whereas Class IV proteins possess a shorter CBD and catalytic region due to deletions [29]. Throughout evolution, several chitinases have lost their ability to hydrolyze or bind chitin [21]. The GH18 group exhibits lower antifungal activity compared to GH19 [30,31] and includes endochitinases that lack a CBD and contain a GH18 catalytic domain. These chitinases are classified into Classes III and V. Class III chitinases feature a C-terminal region and function similarly to lysozymes [26].

Protein chitinases vary in number and mode of action across different species [27,32,33]. Duplication events can alter the genetic composition of chromosomal segments and modify the functions of their gene products [34]. Chitinase families have evolved and expanded through tandem and segmental duplication events, contributing to their high functional diversity [27,33,35]. This diversity is closely related to resistance to biotic stresses and to coevolutionary dynamics between plants and pathogens [36].

Chitinases have been described as a promising source of plant resistance to fungal pathogens through overexpression and genetic transformation [37,38,39,40]. In coffee, fungal diseases such as coffee leaf rust and brown eye spot remain among the main threats to production. Chitinases are directly involved in antifungal defense by degrading the chitin present in fungal cell walls, which makes them particularly relevant for coffee pathosystems. Proteomic analyses have revealed differences in chitinase expression between *C. arabica* cultivars resistant and susceptible to *H. vastatrix* [41,42]. However, the functional characterization and genomic annotation of many of these genes/proteins remain limited. Characterizing gene families, such as chitinases, provides insights into their evolutionary and functional relationships, which can inform genetic improvement programs aimed at enhancing resistance to fungal diseases in coffee. In this context, the current study aimed to characterize the chitinase putative genes and proteins in the genomes of *C. arabica* and its diploid progenitors, *C. canephora* and *C. eugenioides*.

## 2. Results

### 2.1. Identification of Putative Chitinase

To characterize the chitinase putative gene family in coffee, searches were performed using genome-wide HMM profiles of the conserved chitinase domains Glyco_hydro_18 (PF00704) and Glyco_hydro_19 (PF00182), from the Pfam database [43]. A total of 113, 47, and 69 putative chitinase proteins were identified in *C. arabica* (Ca), *C. canephora* (Cc), and *C. eugenioides* (Ce), respectively, with predicted protein lengths ranging from 84 to 788 amino acids (aa). These proteins are encoded by 110 genes in *C. arabica*, 47 in *C. canephora*, and 66 genes in *C. eugenioides*, with three isoforms in *C. arabica* and three in *C. eugenioides*. Domain-based searches revealed that most proteins contained the GH18 catalytic domain (82 in *C. arabica*, 35 in *C. canephora*, and 48 in *C. eugenioides*), whereas a smaller proportion contained the GH19 domain (28, 12, and 18, respectively) (Table 1).

### 2.2. Phylogenetic Analysis

To classify the identified putative chitinase proteins, a maximum likelihood phylogenetic analysis was performed, which separated the proteins from *C. arabica* (*n* = 113), *C. canephora* (*n* = 47), and *C. eugenioides* (*n* = 69) into three main branches. These branches were further categorized into two major groups (GH18 and GH19) and five classes (I, II, III, IV, and V), following the classification established for *Arabidopsis thaliana* [44]. In the GH19 group, a total of 58 proteins were assigned to three classes: Class I, with nine proteins (Ca = 4; Cc = 4; Ce = 1); Class II, with 10 proteins (Ca = 5; Cc = 3; Ce = 2); and Class IV, with 39 proteins (Ca = 19; Cc = 5; Ce = 15) (Figure 1).

The GH18 group contained 171 proteins divided into two classes: Class III, with 115 chitinases (Ca = 57; Cc = 23; Ce = 35) and Class V, with 56 proteins (Ca = 28; Cc = 12; Ce = 16) (Figure 1).

Within GH19, Class I and Class II appear as two related clades in the ML tree (Figure 1). A subset of Class II sequences shows strong bootstrap support, whereas several terminals at the boundary with Class I have moderate support, so their placement is uncertain by phylogeny alone. Because these proteins lack the CBD and, in the orthology analysis, CcChi42, CeChi48, CaChi78, and CaChi81 are grouped with the Class II reference IbChiA, analyses described below, we assigned them to Class II.

### 2.3. Chromosomal Location and Distribution

To investigate the genomic organization of the chitinase family, chromosomal maps were generated for each *Coffea* genome using the start and end positions of the coding genes. For consistency, both genes and their corresponding proteins were named according to the species and chromosomal position: *C. arabica* (CaChi1 to CaChi110); *C. canephora* (CcChi1 to CcChi47) and *C. eugenioides* (CeChi1 to CeChi66). Isoforms were numbered within the same codes (e.g., CaChi37_1 to CaChi37_3; CeChi44_1 to CeChi44_3).

Although *C. arabica* is an allotetraploid species (2*n* = 4*x* = 44), its reference genome is represented by 22 chromosomes due to the high similarity of homoeologous pairs derived from its diploid progenitors, *C. canephora* and *C. eugenioides*, each contributing 11 chromosomes (2*n* = 2*x* = 22). In *C. arabica*, 105 genes were mapped across 18 chromosomes (Chr) of the two subgenomes: 60 genes in the *C. canephora* subgenome and 45 genes in the *C. eugenioides*. Notably, no genes were detected on chromosomes nine and ten of either subgenome, and five genes (CaChi106 to CaChi110) were not annotated as being anchored to any specific chromosome. Genes from the same chitinase classes were found on homologous chromosomes of both subgenomes: Class I (Chr7), Class II (Chr7 and Chr8), Class III (Chr2–5, and Chr11), Class IV (Chr3), and Class V (Chr1, Chr3, Chr6, Chr8) (Figure 2).

In *C. canephora*, 39 genes were distributed across eight of the eleven chromosomes, while eight others were not assigned to any specific chromosome (Figure 3A). In *C. eugenioides*, 64 genes were mapped to all chromosomes except for two that could not be assigned (Figure 3B). In both diploid genomes, chitinase gene classes were predominantly located on homologous chromosomes, although some chromosomes contained genes from more than one class, a pattern also observed in *C. arabica*.

### 2.4. Domain Architecture Analysis and Subcellular Localization

Signal peptides, transmembrane domains, conserved domains, and subcellular localization were analyzed to provide functional insights into the identified proteins. Most sequences exhibited the characteristic chitinase domains GH18 or GH19 when analyzed against the Pfam 35.0 database. However, 14 proteins from *C. arabica* (Appendix A), two from *C. canephora* (Appendix A), and five from *C. eugenioides* (Appendix A) did not show these domains in Pfam. These sequences were reanalyzed in the CDD v.3.21 database and confirmed as members of the GH18_chitinase-like superfamily (cl10447), clustering with Class III chitinases in the phylogenetic analysis.

Proteins containing the Glyco_hydro_19 catalytic domain were grouped into Classes I, II, and IV (Figure 4). Class I proteins generally carried only the GH19 domain, although some (CaChi82, CcChi29, and CeChi49) also contained a chitin-binding domain (CBD) and a signal peptide (SP), resembling the reference protein At3g12500. CeChi49 also carried an extra GH19 domain and a transmembrane domain (TMD), overlapping with the SP, while CaChi84 had a TMD but no CBD. Class II proteins typically contained GH19 and SP domains, except for CcChi42 and CcChi34 which lacked SP. A few Class II proteins (CaChi78, CcChi46, CaChi79, CaChi81, and CeChi48) also showed a TMD overlapping with the SP (Figure 4C).

Most Class IV proteins contained GH19, CBD, and SP domains. Exceptions included CaChi15 (no SP), CeChi54 and CeChi58 (no CBD), CaChi29 and CeChi55 (lacking both). CaChi29 also had an additional GH19 catalytic domain (Figure 4C). Subcellular localization predictions indicated that most GH19 proteins were extracellular, although a few exceptions were detected, such as CaChi15 and CcChi33 (plastids). The subcellular localization of CaChi80 and CcChi30 could not be determined (Figure 4B).

Proteins containing the Glyco_hydro_18 catalytic domain were grouped into Classes III and V (Figure 5A). Most GH18 proteins carried GH18 and SP domains, but several exceptions were observed. Some Class III proteins lacked SP, while others carried additional domains such as Reverse Transcriptase-like 3 domain (RVT-3) (CaChi57), extra GH18 domains (CcChi20, CaChi46, CcChi39), or PKR-like kinase (PKR) (CaChi37_1, CaChi37_3, CaChi38_1, and CeChi20_1) (Figure 5C).

In Class V, many proteins displayed SP overlapping with TMD, only TMD, or only SP. A subset also carried the PKR domain, with variable combinations of SP and/or TMD. Predicted subcellular localization of GH18 chitinases was more diverse than that of GH19, including nucleus, vacuole, extracellular space, plastids, cell membrane, cytoplasm, or Golgi complex. Approximately 58% were predicted to be extracellular, while ~18% had undetermined locations (Figure 5B).

### 2.5. Gene Structure Analysis

The exon-intron structures of *Coffea* spp. chitinases genes were analyzed to assess their structural diversity. Most Class I and II chitinase genes contained three exons, with a few exceptions such as CeChi49 (*n* = 7), CcChi30 (*n* = 1), CcChi31/CcChi33 (*n* = 4). In Class IV, the majority had two or three exons, except for CeChi13 (*n* = 4) and CeChi55 (*n* = 1). In *Coffea arabica*, GH19 chitinase genes typically exhibited the standard structure of two or three exons (Figure 4D).

Class III GH18 chitinase genes were more variable, ranging from one to three exons, with some showing higher counts, such as CaChi55 (*n* = 4), CcChi20 (*n* = 6), and CcChi39 (*n* = 6). Isoform-coding genes, including CaChi37 (*n* = 6), CaChi38 (*n* = 4), and CeChi20 (*n* = 4), also showed exon variability. Similarly, Class V genes usually contained one to three exons, with exceptions such as CeChi40 (*n* = 4) and a cluster in the final clade containing six to eight exons. Many GH18 genes were intronless, in contrast to GH19 where only two intronless genes were detected (Figure 4D and Figure 5D).

### 2.6. Orthologs Analysis

To explore the functional conservation of *Coffea* chitinases, we performed an ortholog analysis including all proteins identified in this study *C. arabica* (*n* = 113), *C. canephora* (*n* = 47), and *C. eugenioides* (*n* = 69), together with 27 reference proteins (QRef) previously described as responsive to fungal pathogen infections. A total of 58 ortholog groups (orthogroups) were identified (Appendix A). Among these, 52 included chitinases from *C. arabica*, 33 from *C. canephora*, 50 from *C. eugenioides*, and nine from reference proteins. Patterns of conservation varied among species. Twenty-four orthogroups contained genes from all three coffee species, while 17 were shared between *C. arabica* and *C. eugenioides*. In contrast, only three orthogroups comprised *C. arabica* and *C. canephora*. Species-specific groups were also found, with three exclusives to *C. eugenioides* and two to *C. arabica*, whereas no groups were exclusive to *C. canephora* or shared only between the two diploid species (Figure 6).

Six orthogroups included chitinases from all three coffee species together with reference proteins, including McChit1, MdCHI1, IbChiA, FvChi-14, CaChitIV, CHI2 and FnCHIT2 (Figure 6 and Figure 7).

### 2.7. Putative Fungal Defense Response Chitinase in Coffea spp.

Based on orthology with reference proteins known to respond to fungal infection, six clusters of *Coffea* chitinases were identified as candidates for defense-related functions. All proteins for which localization could be predicted were assigned to the extracellular region.

Four clusters belonged to the GH19 family. Cluster 1 comprised Class I chitinases located on chromosome 7, ranging from 323 to 584 amino acids (aa). These proteins typically contained SPs and CBDs and three exons; however, CeChi49 was exceptional, with seven exons, a TMD, and an additional GH19 domain. Clusters two and three contained Class II chitinases, characterized by the absence of a CBD and the presence of three exons. Most proteins also possessed an SP domain. Cluster two genes, located on chromosome 7 (except for CcChi42, which was not annotated as anchored on any chromosome), ranged from 194 to 264 aa, and the three proteins of 264 aa carried a TMD. Cluster three genes were located on chromosome 8, mostly encoding proteins of 321 aa with an SP; the exception was CcChi34, with 207 aa and no SP. Cluster IV included Class IV chitinases of 273 aa with two exons, most of which carried SP and CBD. These were chromosome 3, except CeChi56, which was mapped to chromosome 10 and lacked the CBD (Table 2).

Two clusters belonged to the GH18 family. Cluster five comprised Class III chitinases of 294 aa, encoded by single-exon genes and carrying SP. Cluster six included Class V chitinases of 305 aa, also with a single exon and SP (Table 2).

## 3. Discussion

Chitinases are proteins encoded by large multigene families, with individual genes associated with diverse stress-related functions [37]. Numerous studies have focused on their role as pathogenesis-related proteins in different pathosystems, emphasizing expression profiles [26,33,38,39], over-expression studies [50], and the transformation of plants with chitinase genes [38,51] to enhance plant resistance to pathogens. Proteomic analyses revealed a potential role of chitinases in the coffee–fungus interaction [41]. However, no comprehensive characterization of coffee chitinases has been performed. This is the first study to characterize putative chitinase proteins and genes in the genomes of *C. arabica* and its progenitors. Integrating genome-wide analysis, we identified a total of 229 putative chitinase proteins in the three *Coffea* species. Furthermore, based on orthologous analyses, we selected 10, six, and seven putative chitinase genes associated with plant resistance to pathogen infection in *C. arabica*, *C. canephora*, and *C. eugenioides*, respectively.

We obtained a higher number of putative chitinase proteins in *C. arabica* (*n* = 113) compared with its progenitors, *C. canephora* (*n* = 47) and *C. eugenioides* (*n* = 69). This finding is consistent with its allotetraploid status [2], which results in approximately double the genome size of its progenitors [52]. Plant chitinases are generally classified according to two catalytic domains, GH18 and GH19 [21]. Based on the pfam 35.0 database, nearly all chitinases contained one of these domains, with the exception of 14 proteins from *C. arabica*, two from *C. canephora*, and five from *C. eugenioides*. These proteins instead presented the GH18_chitinase-like superfamily (cl10447). Because of the high functional and structural diversity of chitinases, these proteins were retained as candidates for the analyses. Beyond the conserved catalytic domains, several *Coffea* chitinases also displayed features such as signal peptides, transmembrane domains, or additional domains (e.g., kinase or reverse transcriptase-like), indicating structural diversity that may contribute to functional specialization. Phylogenetic analysis further divided the *Coffea* chitinases into two main families (GH18 and GH19) and, in a secondary classification, into five classes (I, II, III, IV, and V), following the previous classification of *A. thaliana* chitinases [44]. Although the majority of plant chitinases have been reported in classes I, II, and IV within the GH19 family [20,22], our results revealed a higher representation of GH18 chitinases in the three *Coffea* genomes.

Chitinase genes in *Coffea* spp. are distributed across various chromosomes and generally occur in tandem, suggesting that tandem duplication has played an important role in their expansion, as observed in other crops such as grape, soybean, tomato, and apple [27,33,38,53]. In *C. arabica*, the subgenome derived from *C. canephora* contained more chitinase genes than the subgenome derived from *C. eugenioides*. Notably, 60 chitinase genes were identified in the *C. canephora*-derived subgenome of *C. arabica*, compared with 47 in the *C. canephora* reference genome. This increase may reflect tandem duplications that occurred after polyploidization or from differences between the *C. canephora* accession used as the reference genome and the one that contributed to the origin of *C. arabica*, or copy number variation already present in the ancestral *C. canephora* donor. This asymmetry supports previously reported subgenome dominance and biased homoeologous exchange favoring the *C. canephora*-derived subgenome, which may be linked to adaptive traits inherited from *C. canephora*, such as its broader ecological range and greater genetic diversity [52]. Consistent with this, in the orthology analysis we observed that *C. eugenioides* often contributed multiple paralogs to orthogroups containing reference chitinases, suggesting lineage-specific duplication or retention within this progenitor.

As expected for *C. arabica*, which is known for its high structural conservation with its progenitors in terms of chromosome number and gene content per chromosome [52], the distribution of chitinase genes was also largely conserved among *C. arabica*, *C. canephora*, and *C. eugenioides*. Most chitinase genes from the same class were located on homologous chromosomes across the three genomes, indicating a conserved gene organization. However, some chromosomes in *C. arabica* contained additional or divergent classes, suggesting post-polyploidization diversification. Furthermore, no chitinase genes were found on chromosomes 9 and 10 of *C. arabica* and *C. canephora,* while *C. eugenioides* showed genes on all chromosomes. This suggests that *C. arabica* may have lost certain chitinases or that these genes were relocated to other chromosomes. Interestingly, most chitinases located on chromosomes 9 and 10 of *C. eugenioides* belonged to Class IV, whereas *C. arabica* contained more Class IV genes on chromosomes 3c and 3e, pointing to potential relocation events. If the genes were simply unannotated, we would expect to find more unanchored Class IV chitinases. However, most unanchored chitinases in all three genomes belong to Class III. Combined with the high genome similarity among the species, this suggests that the absence of some genes is not due to annotation errors. The results indicate that *C. arabica* may have undergone gene loss or gene relocation events. Chitinase genes present in the progenitors and missing in *C. arabica* may be a valuable target for functional studies and genetic improvement, including the introgression of genes associated with enhanced disease resistance in coffee.

In this study, orthologous analysis with previously characterized chitinases associated with enhanced resistance to fungal pathogens enabled the identification of 10, six, and seven candidate genes in *C. arabica*, *C. canephora*, and *C. eugenioides*, respectively. Six orthogroups were identified, each comprising chitinases from all three coffee species and at least one reference gene with known antifungal function. Among these, four genes belonged to the GH19 family and two to GH18. Within GH19, one orthogroup included Class I chitinases that possessed the characteristic chitin-binding domain (CBD) of this class [28]. These chitinases correspond to Mcchit1 and MdCHI1, whose overexpression has been shown to enhance resistance to fungal pathogens in rice and apple, respectively [45,46,47]

Over evolutionary time, some chitinases have lost the ability to hydrolyze or bind to chitin [21], such as the two groups of Class II chitinase proteins, homologous to Class I but lacking the CBD [28]. However, the absence of this domain does not necessarily lead to a loss of enzymatic activity. For example, one of these Class II groups is orthologous to IbChiA, which is highly expressed in resistant sweet potato and exhibits antifungal activity against *Ceratocystis fimbriata* [48]. The other is orthologous to FvChi-14, whose overexpression has been shown to increase *A. thaliana* resistance to *Colletotrichum higginsianum* [39].

A fourth orthogroup included Class IV chitinases associated with a defense-related gene regulator, CaChitIV, from *Capsicum annuum* [49]. Most Class IV chitinases from this orthogroup contained the CBD, except CeChi56. Due to deletions, Class IV chitinases generally have shorter CBD and catalytic regions [29]. In contrast, the coffee GH18 chitinases lacked a CBD, as Classes III and V consist of endochitinases that function similarly to lysozymes, acting through their catalytic domain [26]. Chitinases from these classes were orthologous to CHI2 and FnCHIT2, respectively, both of which have demonstrated critical roles in fungal resistance when overexpressed in transgenic plants [37,40].

All putative chitinases in coffee associated with resistance to fungal pathogens were predicted to be extracellularly localized, except for CeChi2, whose subcellular location could not be determined. This general pattern suggests that these chitinases may function via the secretory pathway, through which many proteins involved in plant–pathogen interactions are released [54]. In addition, most of the corresponding genes contained two or fewer introns, consistent with the typical structure of stress-responsive genes that require rapid transcription [38,55]. An exception is CeChi49, which displayed a more complex exon-intron structure due to a duplicated GH19 domain. Overall, our results support the identification of chitinases associated with fungal defense response in *C. arabica* and its progenitors. These findings provide valuable insights into future research focusing on plant resistance sources to fungal pathogens. The candidate genes identified are orthologous to genes previously used in successful overexpression and transformation studies, supporting their relevance as promising resources for coffee improvement. These candidates can be prioritized for functional studies, and once validated they can be applied in breeding strategies, such as marker-assisted selection and transgenic approaches, to enhance resistance in coffee.

## 4. Materials and Methods

### 4.1. Coffea spp. Genomes

Analysis of chitinase genes and proteins across the genomes of *C. arabica* and its progenitors, *C. canephora* and *C. eugenioides*, involved downloading the FASTA files containing predicted genes and proteins, along with GFF (General Feature Format) annotation files. Genomes used were as follows: *C. arabica* (Caturra red-Cara_1.0, GenBank assembly accession: GCA_003713225.1), *C. eugenioides* (Ceug_1.0, GenBank assembly accession: GCA_003713205.1) from NCBI (National Center for Biotechnology Information), and *C. canephora*, available at NCBI and the Coffee Genome Hub (https://coffee-genome-hub.southgreen.fr/, accessed on 6 October 2025).

### 4.2. In Silico Identification of Putative Chitinase

The HMMER (http://hmmer.org/, accessed on 6 October 2025) software package v. 3.2.1 was used to identify candidate sequences for chitinases in the *C. arabica*, *C. canephora*, and *C. eugenioides* genomes. The Pfam-A.hmm file containing the Hidden Markov Model (HMM) profiles of representative protein families was downloaded from the Pfam database [43]. The function ‘hmmfetch’ was used to retrieve the HMM profiles for the characteristic domains of chitinases, Glyco_hydro_18 (PF00704.31) and Glyco_hydro_19 (PF00182.22). These HMM profiles were subsequently used to search for proteins with the GH18 and GH19 domains in the predicted protein files from *C. arabica*, *C. canephora*, and *C. eugenioides* genomes, using ‘hmmsearch’. Sequences within the threshold (e-value < 10^−4^) were selected for further analysis.

### 4.3. Phylogenetic Analyses

Protein sequences from *Coffea* spp. genomes identified as putative chitinases were aligned with the chitinase proteins of *A. thaliana* described by Passarinho and de Vries, 2002 [44]. Three phylogenetic trees were constructed: one including all sequences identified as belonging to the GH18 and GH19 groups, and two with sequences from each group analyzed separately. Amino acid sequences were aligned using MAFFT v. 6.903 [56,57] with standard parameters. Due to the lack of similarity between the GH18 and GH19 groups, only the separate alignments were filtered using Gblocks 0.91b [58] to detect the most conserved regions. MAFFT and Gblocks were executed on NGPhylogeny.fr (https://ngphylogeny.fr/, accessed on 6 October 2025) [59]. Trees were inferred using IQ-TREE (http://iqtree.cibiv.univie.ac.at/, accessed on 6 October 2025) [60] with the Maximum Likelihood (ML) method and 1000 bootstrap replicates. The best-fit model, WAG + G4, was selected for all phylogenetic trees based on the Bayesian information criterion (BIC). The phylogenetic trees were visualized and edited using the Interactive Tree of Life (iTOL) [61]. Chitinase proteins from *A. thaliana* were used as outgroups in the phylogenetic trees: At1g02360 for the GH18 + GH19 concatenated and individual GH19 trees, and At4g19720 for the individual GH18 tree.

### 4.4. Chromosomal Mapping of Chitinase Genes

Chromosomal location data for chitinase-coding genes (GFF annotation file) and chromosome sizes in Mb for the *C. arabica*, *C. canephora*, and *C. eugenioides* genomes were collected from the NCBI database and the Coffee Genome Hub. Chromosomal mapping and schematic visualization of chitinase genes were performed using MapGene2Chromosome v.2.1 (MG2C, http://mg2c.iask.in/mg2c_v2.1/, accessed on 6 October 2025). Gene position data (start and end coordinates), chromosome numbers, chromosome lengths, and specific colors were provided in the format required by MG2C.

### 4.5. Domain Architecture Analysis and Subcellular Location

Protein sequences identified as chitinases based on GH18 and GH19 HMM profiles and *Arabidopsis thaliana* references [44] were validated for domain presence using NCBI’s Conserved Domain Database (CDD) [62] with Pfam v.35.0. Sequences that lacked domain matches in Pfam were reanalyzed against the CDD v.3.21 database using the same default parameters, including an E-value threshold of 0.01, composition-based statistics adjustment, and a maximum of 500 hits, with result mode set to “Concise”. Signal peptides were predicted with SignalP 6.0 [63], and transmembrane domains with TMHMM 2.0 [64]. The subcellular localization was predicted using Plant-mSubP (http://bioinfo.usu.edu/Plant-mSubP/, accessed on 6 October 2025) [65] with the four available prediction methodologies (AAC, Dipep, PseAACNCCDipep, and NCCDipepCTDCCTDTQSO). The most frequent prediction was considered (Appendix A). The localization was classified as undetermined when each method yielded a different outcome. All tools were used with their default confidence score thresholds.

### 4.6. Gene Structure Annotation and Analysis

Gene structure was examined using the Gene Structure Display Server (GSDS) [66], based on CDS and corresponding genomic DNA sequences provided in FASTA format through the “Sequence FASTA” option. CDS sequences were retrieved from NCBI (https://www.ncbi.nlm.nih.gov/, accessed on 6 October 2025) by searching for protein IDs, from which the associated gene IDs and genomic sequences were obtained. For *Coffea canephora*, genomic sequences were not available in NCBI. Thus, protein sequences were aligned against the Coffee Genome Hub (https://coffee-genome-hub.southgreen.fr/, accessed on 6 October 2025) using BLAST, and the corresponding genomic regions were identified based on matching gene IDs. Domain architecture, gene structure, and subcellular localization were annotated and mapped onto the phylogenetic tree using the Interactive Tree of Life (iTOL) v.7.2.2, with visual features customized through the iTOL annotation editor [61]. Final figures were edited using Inkscape v.1.3.2 (https://inkscape.org, accessed on 6 October 2025).

### 4.7. Orthogroups Analysis

The protein sequences of candidate chitinases from the genomes of three coffee species were analyzed alongside functionally characterized plant chitinases, which are known to respond to fungal diseases based on a literature review of Chen et al., 2024 [67] and are here referred to as reference chitinases (QRef) (Appendix A). These reference-sequences correspond to chitinases used in genetic transformation or overexpression experiments that improved plant resistance to fungal pathogens. The FASTA sequences of QRef were retrieved from the NCBI database or from other databases specified in the consulted articles. We then performed the orthologous analysis using OrthoVenn3 [68] with the default OrthoMCL algorithm parameters, including an E-value of 10^−2^ and an inflation value of 1.5. Annotation, protein similarity, and cluster relationship network options were enabled.

## 5. Conclusions

Plant chitinases exhibit high diversity and play key roles in growth, development, and responses to biotic and abiotic stress. Identifying and characterizing these putative genes and proteins is essential for their application in biotechnology. This study provides the first comprehensive characterization of chitinase genes and proteins in the genome of *Coffea arabica* and its diploid progenitors, *C. canephora* and *C. eugenioides*, based on phylogenetic relationships, domain architecture, gene structure, and subcellular localization. Characterizing the progenitor genomes expands the available gene pool and helps overcome the complexity of the allotetraploid *C. arabica* genome. We found that chitinases are highly conserved across the three genomes, supporting the potential use of genes from the progenitor species in *C. arabica* through genetic transformation. Furthermore, ortholog analysis enabled the identification of putative chitinases involved in plant–pathogen interactions, which could be explored not only in coffee but also in other crops through gene overexpression, the development of biobased fungicides, and the generation of transgenic plants. Altogether, these findings provide a valuable foundation for future breeding and biotechnological strategies aimed at improving fungal disease resistance.

## Figures and Tables

**Figure 1 plants-14-03130-f001:**
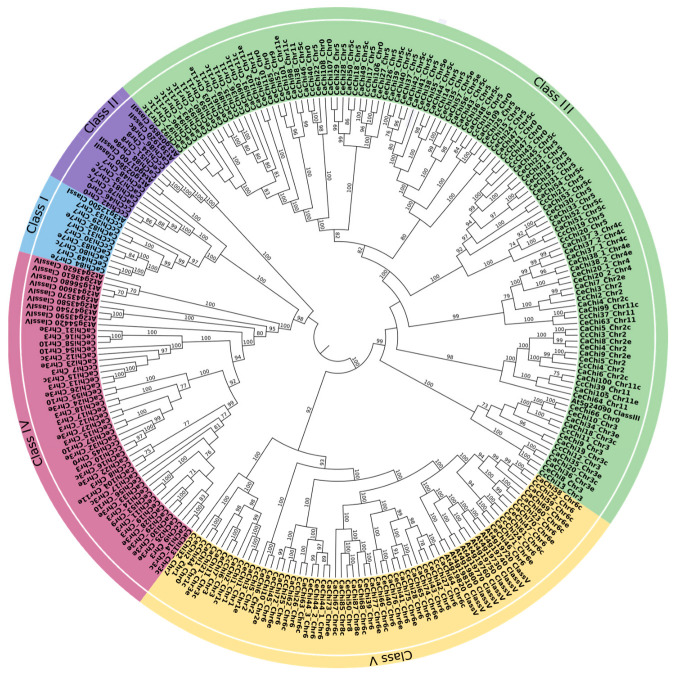
Maximum likelihood phylogenetic tree constructed with putative chitinase proteins from *Arabidopsis thaliana*, *Coffea arabica*, *Coffea canephora*, and *Coffea eugenioides*. Proteins were classified into five major classes according to their glycosyl hydrolase families: GH19 (Classes I, II, and IV) and GH18 (Classes III and V), represented in different colors. Bootstrap support values ≥ 70% are shown at the nodes. The *A. thaliana* protein At1g02360 was used as the outgroup.

**Figure 2 plants-14-03130-f002:**
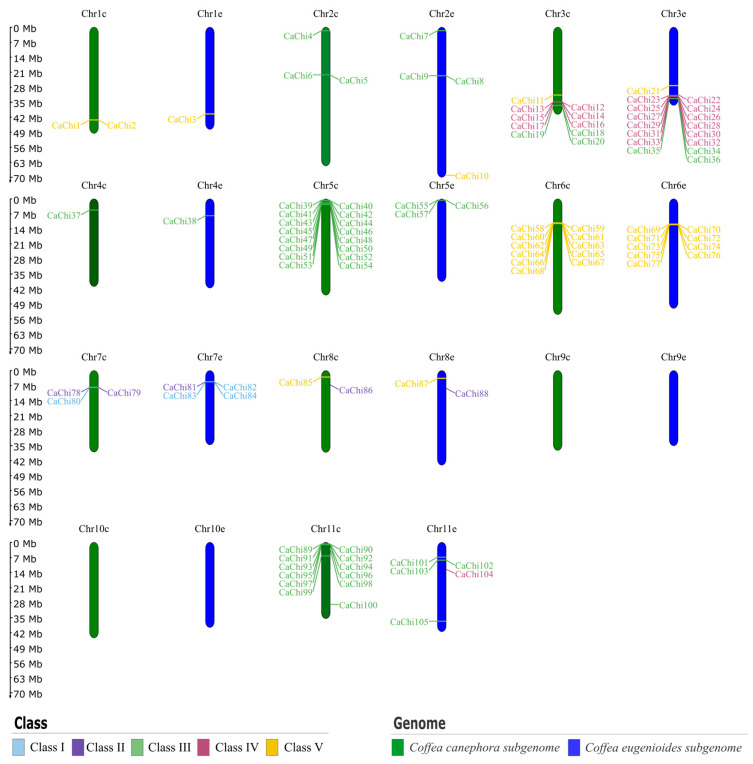
Chromosomal location of chitinase genes in *Coffea arabica*. The complete set of 22 chromosomes (2*n* = 44) is shown, representing both parental subgenomes. The prefix “Chr” indicates chromosome. Chromosomes labeled with “c” (e.g., Chr1c) refer to the *C. canephora* subgenome, and those with “e” (e.g., Chr1e) to the *C. eugenioides* subgenome. Individual chitinase genes are indicated on each chromosome, with gene classes distinguished by color coding.

**Figure 3 plants-14-03130-f003:**
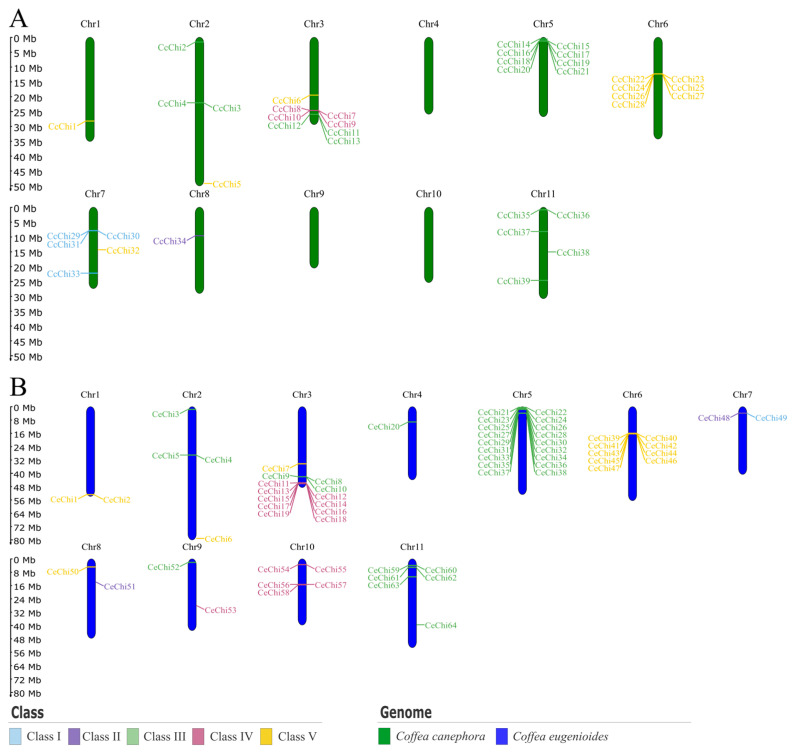
Chromosomal location of chitinase genes in *Coffea* spp. (**A**) *Coffea canephora* and (**B**) *Coffea eugenioides*. Each panel shows the complete chromosome set of the respective diploid species (2*n* = 22), with chitinase genes mapped to their physical positions. Chromosomes are labeled with the prefix “Chr” followed by their number. Gene classes are color-coded, allowing the visualization of how different chitinase groups are distributed across chromosomes.

**Figure 4 plants-14-03130-f004:**
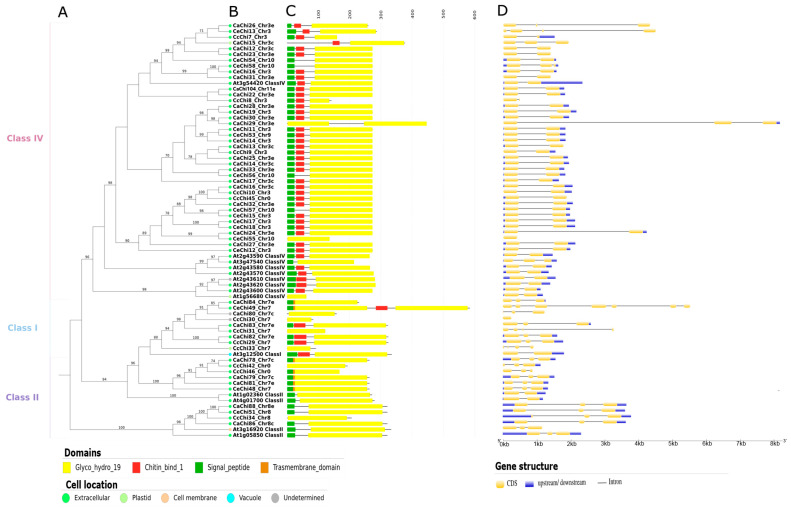
Characterization of putative GH19 chitinases from *Arabidopsis thaliana*, *Coffea arabica*, *Coffea canephora*, and *Coffea eugenioides*. (**A**) Maximum likelihood phylogenetic tree with bootstrap support values ≥ 70% shown at the nodes, using *A. thaliana* At1g02360 as the outgroup; (**B**) Predicted subcellular localization, with different colors representing distinct cellular compartments; (**C**) Conserved domain structure of the chitinase proteins; (**D**) exon-intron structure of the genes, with exons shown in yellow and introns represented by black lines.

**Figure 5 plants-14-03130-f005:**
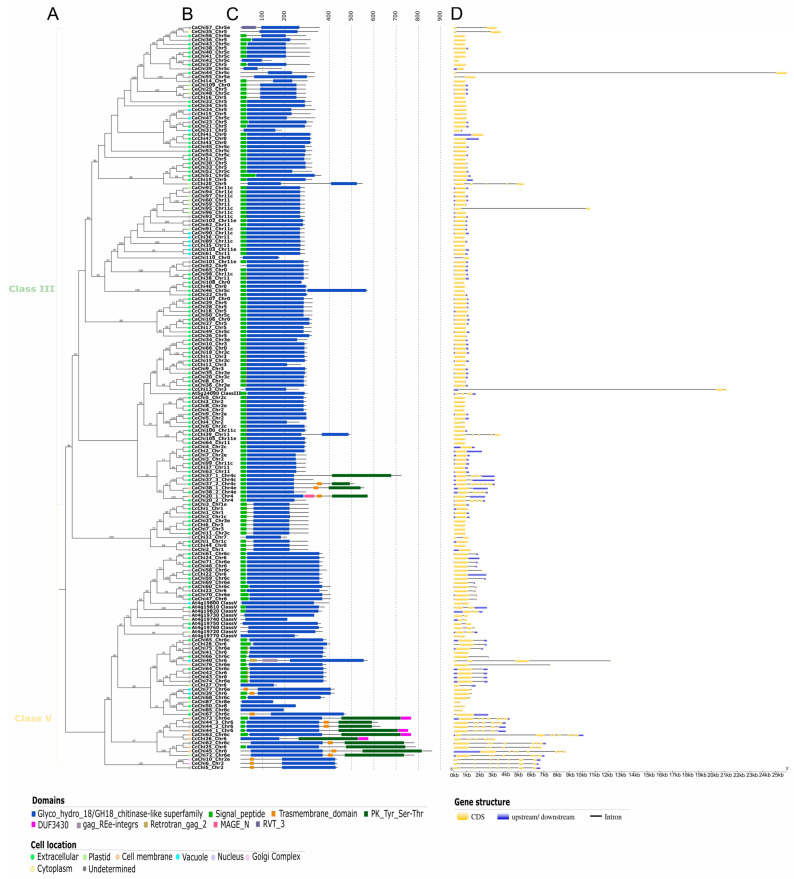
Characterization of putative GH18 chitinases from *Arabidopsis thaliana*, *Coffea arabica*, *Coffea canephora*, and *Coffea eugenioides*. (**A**) Maximum likelihood phylogenetic tree with bootstrap support values ≥ 70% shown at the nodes, using *A. thaliana* At4g19720 as the outgroup; (**B**) Predicted subcellular localization, with different colors representing distinct cellular compartments; (**C**) Conserved domain structure of the chitinase proteins; (**D**) exon-intron structure of the genes, with exons shown in yellow and introns represented by black lines.

**Figure 6 plants-14-03130-f006:**
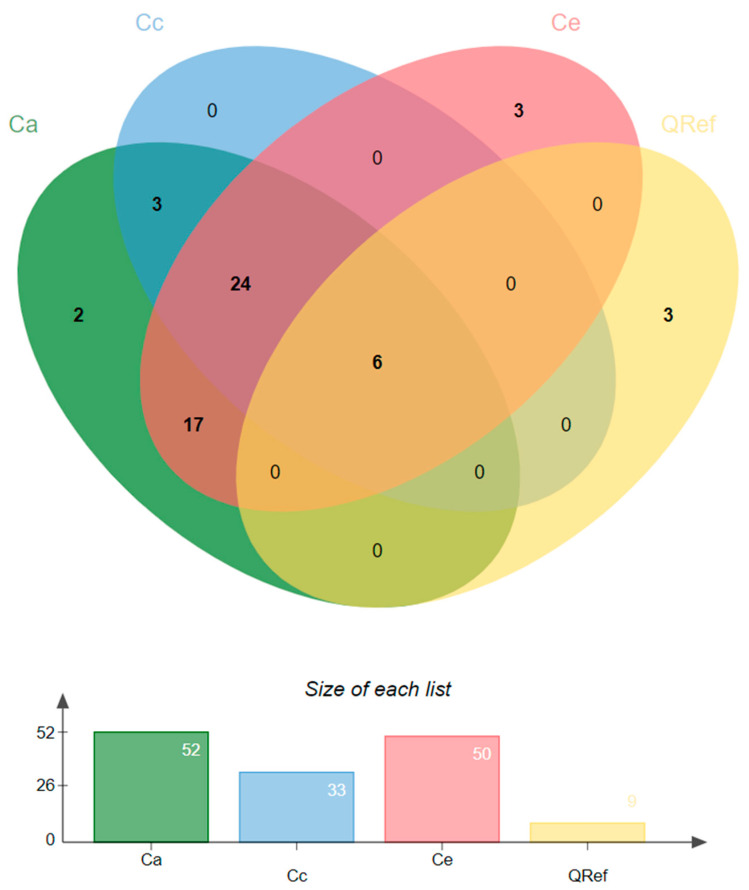
Orthologous groups of chitinases (orthogroups) among the three coffee species, *Coffea arabica* (Ca), *Coffea canephora* (Cc), *Coffea eugenioides* (Ce), and reference chitinases (QRef). The Venn diagram displays shared and unique orthogroups across species. The “Size of each list” indicates the number of orthogroups assigned to each species or reference sequence.

**Figure 7 plants-14-03130-f007:**
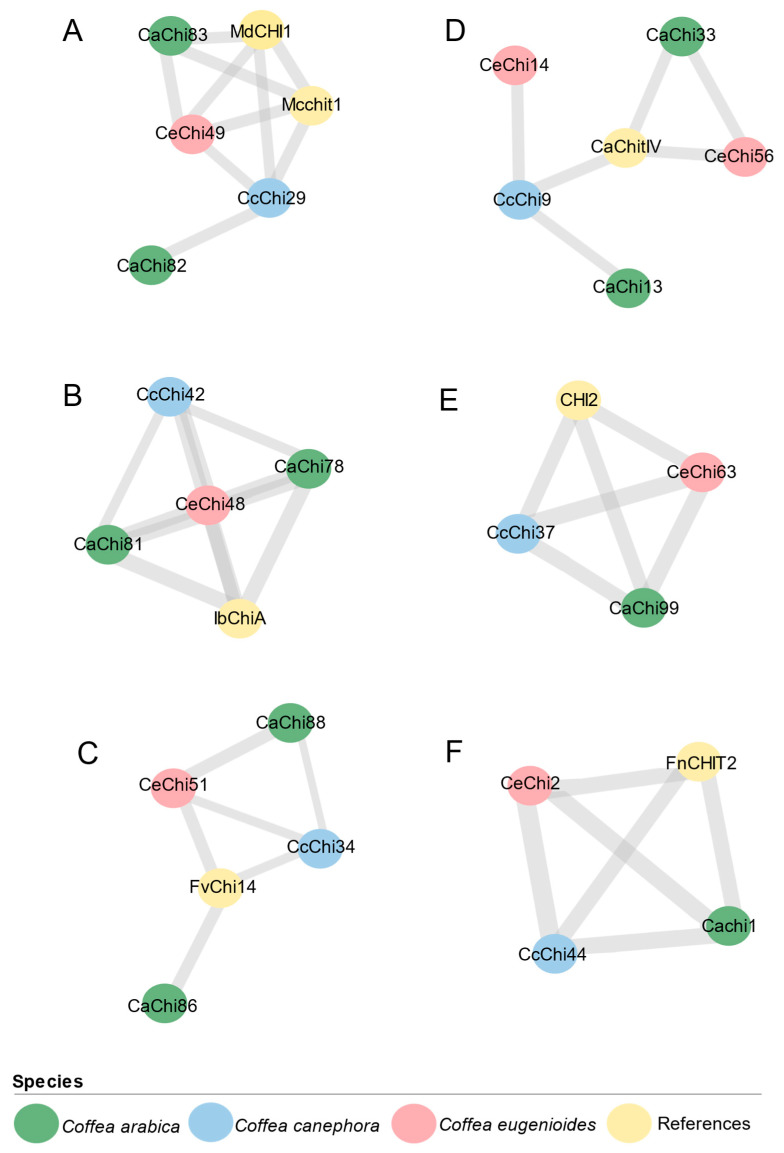
Orthologous Clusters grouping putative chitinases from *Coffea arabica* (CaChi), *Coffea canephora* (CcChi), and *Coffea eugenioides* (CeChi) to chitinase references. (**A**) Chitinases Class I clustered with references MdCHI1 and Mcchit1; (**B**) Chitinases Class II clustered with reference IbCHIA; (**C**) Chitinases Class II clustered with reference FvChi-14; (**D**) Chitinases Class IV clustered with reference CaChitIV; (**E**) Chitinases Class III clustered with reference CHI2; (**F**) Chitinases Class V clustered with reference FnCHIT2. Node colors represent different species. Line thickness indicates the degree of sequence similarity between proteins.

**Table 1 plants-14-03130-t001:** Putative chitinase proteins: number of proteins, protein size range, number of coding genes, and number of coding genes classified into GH18 and GH19 groups.

Coffee Species	Proteins	Proteins Size	Genes	GH18 Genes	GH19 Genes
*Coffea arabica*	113	141–782	110	82	28
*Coffea canephora*	47	84–788	47	35	12
*Coffea eugenioides*	69	187–862	66	48	18

**Table 2 plants-14-03130-t002:** Characterization summary of putative chitinases associated with defense response to fungi in *Coffea arabica*, *Coffea canephora*, and *Coffea eugenioides*.

Cluster	Reference	Protein	Class	Size	Chr	Cell Localization	Domains ^1^				Exons
1	Mcchit1 [45,46] MdCHI1 [47]	CcChi29	I	324	7	Extracellular	SP		CBD	GH19	3
1	Mcchit1 [45,46] MdCHI1 [47]	CeChi49	I	584	7	Extracellular	SP	TMD	CBD	2 GH19	7
1	Mcchit1 [45,46] MdCHI1 [47]	CaChi83	I	323	7e	Extracellular	SP		CBD	GH19	3
1	Mcchit1 [45,46] MdCHI1 [47]	CaChi82	I	324	7e	Extracellular	SP		CBD	GH19	3
2	IbChiA [48]	CaChi81	II	264	7e	Extracellular	SP	TMD		GH19	3
2	IbChiA [48]	CaChi78	II	264	7c	Extracellular	SP	TMD		GH19	3
2	IbChiA [48]	CcChi42	II	194	0	Extracellular				GH19	3
2	IbChiA [48]	CeChi48	II	264	7	Extracellular	SP	TMD		GH19	3
3	FvChi-14 [39]	CcChi34	II	207	8	Extracellular				GH19	3
3	FvChi-14 [39]	CeChi51	II	321	8	Extracellular	SP			GH19	3
3	FvChi-14 [39]	CaChi86	II	321	8c	Extracellular	SP			GH19	3
3	FvChi-14 [39]	CaChi88	II	321	8e	Extracellular	SP			GH19	3
4	CaChitIV [49]	CcChi9	IV	273	3	Extracellular	SP		CBD	GH19	2
4	CaChitIV [49]	CeChi56	IV	273	10	Extracellular	SP			GH19	2
4	CaChitIV [49]	CaChi33	IV	273	3e	Extracellular	SP		CBD	GH19	2
4	CaChitIV [49]	CeChi14	IV	273	3	Extracellular	SP		CBD	GH19	2
4	CaChitIV [49]	CaChi13	IV	273	3c	Extracellular	SP		CBD	GH19	2
5	CHI2 [40]	CcChi37	III	294	11	Extracellular	SP			GH18	1
5	CHI2 [40]	CeChi63	III	294	11	Extracellular	SP			GH18	1
5	CHI2 [40]	CaChi99	III	294	11c	Extracellular	SP			GH18	1
6	FnCHIT2 [37]	CcChi44	V	305	0	Extracellular	SP			GH18	1
6	FnCHIT2 [37]	CeChi2	V	305	1	Undetermined	SP			GH18	1
6	FnCHIT2 [37]	CaChi1	V	305	1c	Extracellular	SP			GH18	1

^1^ SP: Signal Peptide; TMD: Transmembrane domain; GH19: Glyco_hydro19; GH18: Glyco_hydro18; CBD: Chitin-binding domain.

## Data Availability

Data are contained within the article and Appendix A.

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
