# Peer review of "Genome-Wide Analysis Reveals Chitinases as Putative Defense-Related Proteins Against Fungi in the Genomes of Coffea arabica and Its Progenitors"

_plants, 2025, doi:10.3390/plants14203130_

Round 1

Reviewer 1 Report

Comments and Suggestions for Authors

This study identifies and characterizes chitinase genes in Coffea arabicaC. canephora, and C. eugenioides using phylogenetic, structural, and orthology analyses. A total of 113, 47, and 69 putative chitinases were classified into GH18/GH19 families and Classes I-V. Ten, six, and seven candidate chitinases linked to fungal defense were identified, providing valuable insights for future research on coffee pathogen resistance. The work is well-structured but could benefit from experimental validation of chitinase functions.

I believe the following issues need to be addressed:

  1. Research Background – Is the logical connection between chitinases and coffee sufficiently justified?
  2. Discussion Section – While mechanistic explanations are thorough, does the study provide practical implications for coffee breeding?
  3. References – Some formatting inconsistencies require careful verification.
  4. Scientific Nomenclature – Certain Latin species names and gene names are not italicized.
  5. Figure Legends – Several legends are overly simplistic and should be expanded for clarity.

Author Response

Major comment: This study identifies and characterizes chitinase genes in Coffea arabica, C. canephora, and C. eugenioides using phylogenetic, structural, and orthology analyses. A total of 113, 47, and 69 putative chitinases were classified into GH18/GH19 families and Classes I–V. Ten, six, and seven candidate chitinases linked to fungal defense were identified, providing valuable insights for future research on coffee pathogen resistance. The work is well-structured but could benefit from experimental validation of chitinase functions.

Response: We thank the reviewer for this important comment. We agree that experimental validation is essential to confirm the functions of chitinases in coffee defense. The main objective of our study was to provide a comprehensive identification and classification of chitinase genes, since they represent a large and diverse family with roles in antifungal defense, plant development, and stress responses. Establishing this genomic framework is a necessary first step before detailed functional studies can be performed. Based on this foundation, ongoing studies in our group are already focusing on validating the candidate genes through expression analysis under fungal infection and functional assays in coffee plants. We believe this combined approach will provide stronger insights into the role of chitinases in coffee resistance.

Comments 1: Research Background – Is the logical connection between chitinases and coffee sufficiently justified?

Response 1: Thank you for this comment. We revised the Introduction to better emphasize the connection between chitinases and coffee, highlighting their role in defense against fungal pathogens and their potential use in improving disease resistance in coffee

Comments 2: Discussion Section – While mechanistic explanations are thorough, does the study provide practical implications for coffee breeding?

Response 2: Thank you for this comment. We added a sentence at the end of the Discussion to highlight the practical implications, noting that the candidate chitinase genes can be prioritized in functional studies and, once validated, applied in breeding and biotechnological approaches to improve resistance in coffee.

Comments 3: References – Some formatting inconsistencies require careful verification.

Response 3: Thank you for this comment. The references have been carefully reviewed and corrected for formatting consistency.

Comments 4: Scientific Nomenclature – Certain Latin species names and gene names are not italicized.

Response 4: Thank you for this comment. The manuscript has been carefully revised, and all Latin species names and gene names have been corrected to italics.

Comments 5: Figure Legends – Several legends are overly simplistic and should be expanded for clarity.

Response 5: Thank you for this comment. The figure legends have been revised and expanded to provide greater clarity.

Reviewer 2 Report

Comments and Suggestions for Authors

The present study performs a complete genome-wide analysis of chitinase genes in Coffea arabica and its progenitors. From the bioinformatics perspective, it is a valuable study, as it integrates phylogenetic, structural, and functional annotations and provides valuable insights into gene evolution and potential roles in fungal defense. But it has a major problem:

There is no evidence that any of the described genes is functional; this greatly limits the applicability of the results and the value of the paper. There is also no analysis of the promoters, so the reader has no indication of whether the described genes are indeed real genes or pseudogenes. So I recommend including evidence of to what extent the genes are expressed or not. A major point would be to include information on whether the genes are upregulated by fungal attack. Also the localization is based on structural prediction. So the paper lacks some "wet" experiments to confirm the bioinformatics predictions.

Minor points:

The manuscript's organization could be improved for clarity, particularly in the Results and Discussion sections, where some findings are scattered and sometimes lack contextual explanation. Also, there are some inconsistencies in gene nomenclature and classification across different sections, which may cause confusion.

Figures 2 and 3 cannot be read. Enhance resolution and consider the use of darker colors.

Figures 4 and 5: try to enlarge, as the lettering cannot be read.

Please check for typographical issues and formatting inconsistencies.

Author Response

Major Comment: The present study performs a complete genome-wide analysis of chitinase genes in Coffea arabica and its progenitors. From the bioinformatics perspective, it is a valuable study, as it integrates phylogenetic, structural, and functional annotations and provides valuable insights into gene evolution and potential roles in fungal defense. But it has a major problem: There is no evidence that any of the described genes is functional; this greatly limits the applicability of the results and the value of the paper. There is also no analysis of the promoters, so the reader has no indication of whether the described genes are indeed real genes or pseudogenes. So I recommend including evidence of to what extent the genes are expressed or not. A major point would be to include information on whether the genes are upregulated by fungal attack. Also the localization is based on structural prediction. So the paper lacks some "wet" experiments to confirm the bioinformatics predictions.

Response: Thank you for this important comment. We agree that promoter analysis, expression profiling, and functional validation are very important to confirm the role of these chitinases in coffee defense. The main goal of our study, however, was to provide the first complete genomic framework of the chitinase family in C. arabica and its progenitors. This framework is the necessary first step, and only with it will it be possible to select genes for promoter studies, expression analysis, and functional validation. Based on this study, our group is now working on expression studies under fungal infection and functional assays in coffee plants.

Comments 1: The manuscript's organization could be improved for clarity, particularly in the Results and Discussion sections, where some findings are scattered and sometimes lack contextual explanation. Also, there are some inconsistencies in gene nomenclature and classification across different sections, which may cause confusion.

Response 1: Thank you. We carefully revised the Results and Discussion sections to improve clarity and flow, and we corrected inconsistencies in gene nomenclature and classification to ensure uniformity throughout the manuscript.

Comments 2: Figures 2 and 3 cannot be read. Enhance resolution and consider the use of darker colors.

Response 2: Thank you for this observation. Figures 2 and 3 were generated at 300 dpi resolution and provided in multiple formats (SVG, TIFF, and PDF) to ensure quality during the publication process. We kept the same color scheme as in Figure 1 to maintain consistency across figures.

Comments 3: Figures 4 and 5: try to enlarge, as the lettering cannot be read.

Response 3: Thank you. All textual elements in Figures 4 and 5 have been enlarged, and the figures have been prepared at high quality and saved in multiple formats (PNG,SVG, and PDF) at 300 dpi. While space limitations in the Word document may cause the lettering to appear smaller, the figures retain excellent resolution. By zooming in, all elements can be clearly read, and the quality remains fully preserved

Comments 4: Please check for typographical issues and formatting inconsistencies.

Response 4: Thank you. The manuscript has been carefully revised, and typographical and formatting inconsistencies have been corrected.

Reviewer 3 Report

Comments and Suggestions for Authors

Sections 2.2 and 2.4 have to be corrected according to my comments in attached file.

As some results are not commented in Discussion section it would be fine at least mention that such analysis were done.

My other comments are marked in attached file.

Author Response

Major Comment: Sections 2.2 and 2.4 have to be corrected according to my comments in attached file. As some results are not commented in Discussion section it would be fine at least mention that such analysis were done. My other comments are marked in attached file.

Response: Thank you for these important comments. Sections 2.2 and 2.4 have been revised according to your suggestions. We also updated the Discussion to mention and briefly comment on analyses that were previously not addressed. In addition, we carefully revised the manuscript to incorporate all other corrections indicated in the attached file.

Comments 1: Latin names should be in italics both here and in the whole text

Response 1: Thank you. All Latin names have been carefully checked and corrected to italics throughout the entire manuscript.

Comments 2: Coffee

Response 2: Thank you. It has been corrected for coffee species.

Comments 3: Judging from the former section and table, the GH18 group has 165 members. How can there be 115 + 56 = 171 chitinases?

Response 3: Thank you for this observation. The difference is because in the Results section we report proteins, which include isoforms, while in Table 1 we present the number of genes. Therefore, the GH18 group has 165 genes, but these correspond to 171 proteins. To make this clearer, we specified “genes” in the table.

Comments 4: I can see 15.

Response 4: Thank you for your comment. During the revision process, this section was rewritten, and the information regarding the number of observations was removed from the text.

Comments 5: I can see only six of them.

Response 5: Thank you for this comment. There are indeed ten proteins with TMD overlapping the SP. Because the two domains have the same size and overlap in the figure, some of them are difficult to see. By zooming into Figure 5, especially in the PDF version with higher quality, it is possible to see the SP domain in green and a small part of the TMD domain in orange for the first four proteins of Class V.

Comments 6: CcChi42 which has 194 aa is not on chromosome 7.

Response 6: Thank you for this observation. You are correct, and we have corrected this in the text.

Comments 7: Type the Table with smaller font avoiding splitting like: Cluste r/ Extracellula r.

Response 7: Thank you for this suggestion. The table has been reformatted with a smaller font to avoid word splitting.

Comments 8: As the numbers are also given in line 315 it may be better to give the sum of them (without "respectively") in this sentence?

Response 8: Thank you for this suggestion. We agree and have modified the sentence to present the total number of putative chitinases instead of listing them individually with “respectively.”

Comments 9: This was not mentioned before in Results section.

Response 9: Thank you for this comment. We would like to clarify that this information was already included in the Results section (lines 182-185). In this section, we reported that 14 proteins from C. arabica, two from C. canephora, and five from C. eugenioides did not contain the characteristic chitinase domains.

Reviewer 4 Report

Comments and Suggestions for Authors

In the manuscript named “Genome-wide analysis reveals chitinases as putative fungal defense responses in the genomes of Coffea arabica and its progenitors”, authors have performed genome-wide analysis of chitinases genes in Coffea arabica and its progenitors. They have characterized with phylogenetic analysis, domain architecture, subcellular localization of these genes, and they have also investigated their potential roles in response to fungal stress. These findings would be valuable for coffee genetic improving in future. However, there were some comments about it.

(1) Authors have performed genome-wide analysis of chitinases genes in some coffee genome, but all works were bioinformatics analysis, no molecular evidence was present in this manuscript, such as qRT-PCR, or GFP, etc.

(2) Authors have characterized chitinases with potential roles in fungal defense responses, based blast against previous chitinases [ref70], but no direct evidence was supporting, from gene expression to transgenic, even other bioinformatics analysis. The conclusion was poorly supported by present results, please consider more evidence.

(3) “It identified 58 ortholog groups (Table S4), of which 52 included chitinases from C. arabica, 33 from C. canephora, 50 from C. eugenioides and nine from the reference chitinase proteins” from line 248 to 250. The results had shown most chitinases from C. eugenioides were orthologs of refs, the ratio was extremely high, it indicated few members of chitinases had many repeat events in C. eugenioides genome. It would be explained in detail.

(4) The authors had compared three genomes, it was recommended that they conducted a synteny analysis of these three genomes, particularly given that some chitinases members exhibiting substantial differences in their partial distribution trends across the chromosomes of the three genomes.

(5) In Coffea canephora genome, there was some protein with 84aa, which was smaller than many members, please check them.

(6) In hmmsearch, authors had selected “e-value <10-4”, why? Not 0.01?

Author Response

Major Comment: In the manuscript named “Genome-wide analysis reveals chitinases as putative fungal defense responses in the genomes of Coffea arabica and its progenitors”, authors have performed genome-wide analysis of chitinases genes in C. arabica and its progenitors. They have characterized with phylogenetic analysis, domain architecture, subcellular localization of these genes, and they have also investigated their potential roles in response to fungal stress. These findings would be valuable for coffee genetic improving in future. However, there were some comments about it.

Response: Thank you for recognizing the value of our study for coffee improvement. We revised the manuscript carefully and addressed all the specific comments to improve clarity.

Comments 1: Authors have performed genome-wide analysis of chitinase genes in some coffee genomes, but all works were bioinformatics analysis, no molecular evidence was present in this manuscript, such as qRT-PCR, or GFP, etc.

Response 1: Thank you for this comment. We agree that experimental validation is important. The goal of this work was to build the first genomic framework of the chitinase family in coffee, which is a necessary first step before qRT-PCR or functional studies can be done. Based on this study, our group is now continuing with expression and functional studies.

Comments 2: Authors have characterized chitinases with potential roles in fungal defense responses, based on BLAST against previous chitinases [ref70], but no direct evidence was supporting, from gene expression to transgenic, even other bioinformatics analysis. The conclusion was poorly supported by present results, please consider more evidence.

Response 2: Thank you for your comment. We agree that additional evidence such as expression data or transgenic validation would indeed be valuable to confirm the role of these chitinases in fungal defense. However, the aim of this study was limited to providing the first genome-wide framework of the chitinase family in coffee, integrating phylogenetic, structural, and orthology analyses. This genomic foundation is a necessary step before detailed functional studies can be pursued, but such experiments are beyond the scope of the present work.

Comments 3: “It identified 58 ortholog groups (Table S4), of which 52 included chitinases from C. arabica, 33 from C. canephora, 50 from C. eugenioides and nine from the reference chitinase proteins” from line 248 to 250. The results had shown most chitinases from C. eugenioides were orthologs of refs, the ratio was extremely high, it indicated few members of chitinases had many repeat events in C. eugenioides genome. It would be explained in detail.

Response 3: Thank you for this helpful suggestion. In the orthology analysis, we observed that C. eugenioides frequently contributed multiple paralogs to orthogroups containing reference chitinases, which may reflect lineage-specific duplication or retention events. We have now added this point to the Discussion (lines 446-448) to better highlight this observation.

Comments 4: The authors had compared three genomes, it was recommended that they conducted a synteny analysis of these three genomes, particularly given that some chitinases members exhibiting substantial differences in their partial distribution trends across the chromosomes of the three genomes.

Response 4: Thank you for this suggestion. We agree that a synteny analysis could provide additional insights, but it was outside the main scope of this study. Our main goal was to identify and characterize the chitinase family in coffee and to highlight candidate genes potentially involved in fungal defense, which will guide the continuation of this work.

Comments 5: In Coffea canephora genome, there was some protein with 84 aa, which was smaller than many members, please check them.

Response 5: Thank you for this observation. We agree that some coffee chitinases are shorter than typical members of the family. As our goal was to capture the full diversity of the gene family, we retained all sequences that presented GH18 or GH19 catalytic domains, regardless of their length.

Comments 6: In hmmsearch, authors had selected “e-value <10⁻⁴”, why? Not 0.01?

Response 6: Thank you for this observation. We applied an e-value cutoff of 10⁻⁴ to minimize false positives. This threshold has also been commonly used in similar genome-wide chitinase studies, ensuring a balance between sensitivity and specificity, for example:

Genome-Wide Identification of the Maize Chitinase Gene Family and Analysis of Its Response to Biotic and Abiotic Stresses, https://doi.org/10.3390/genes15101327.

Genome-wide identification and expression profiling of chitinase genes in tea (Camellia sinensis L.) O. Kuntze) under biotic stress conditions, https://doi.org/10.1007/s12298-021-00947-x.

Genome-Wide Identification and Expression Analyses of Glycoside Hydrolase Family 18 Genes During Nodule Symbiosis in Glycine max, https://doi.org/10.3390/ijms26041649.

Analysis of chitinase gene family in barley and function study of HvChi22 involved in drought tolerance, https://doi.org/10.1007/s11033-024-09651-x.

Characterization of soybean chitinase genes induced by rhizobacteria involved in the defense against Fusarium oxysporum, https://doi.org/10.3389/fpls.2024.1341181.

Round 2

Reviewer 2 Report

Comments and Suggestions for Authors

The revised version has improved the formal aspects, but the major problem remains. We do not know to what extent the information is relevant. This paper is lacking a mere rtPCR analysis to sort out which genes are really expressed and which are pseudogenes. Without this I cannot endorse publication. 

Author Response

Comment: The revised version has improved the formal aspects, but the major problem remains. We do not know to what extent the information is relevant. This paper is lacking a mere rtPCR analysis to sort out which genes are really expressed and which are pseudogenes. Without this I cannot endorse publication. 

Response: We thank the reviewer for the additional comment. However, we would like to reiterate that the scope of our study is restricted to a genome-wide bioinformatic analysis. Experimental expression data, such as RT-PCR, while valuable, are beyond the objectives of this work. All points requested by the Editor have been fully addressed in the revised version.

Reviewer 4 Report

Comments and Suggestions for Authors

Thanks for authors’ works, the manuscript had been well revised, but the comments #1 and #2 were still here. There was no molecular experiment to support their conclusion. Therefore, a decision from the editor is needed regarding whether the manuscript can be published. Good luck.

Author Response

Comment: Thanks for authors’ works, the manuscript had been well revised, but the comments #1 and #2 were still here. There was no molecular experiment to support their conclusion. Therefore, a decision from the editor is needed regarding whether the manuscript can be published. Good luck.

Response: We thank the reviewer for the feedback and for acknowledging the improvements in our revision. As noted, the scope of this study is limited to bioinformatic analyses, and all points requested by the Editor have been fully addressed.